# Chemotherapy-Induced Intestinal Microbiota Dysbiosis Impairs Mucosal Homeostasis by Modulating Toll-like Receptor Signaling Pathways

**DOI:** 10.3390/ijms22179474

**Published:** 2021-08-31

**Authors:** Ling Wei, Xue-Sen Wen, Cory J. Xian

**Affiliations:** 1School of Pharmaceutical Sciences, Cheeloo College of Medicine, Shandong University, Jinan 250012, China; WL0305@mail.sdu.edu.cn; 2UniSA Clinical & Health Science, City West Campus, University of South Australia, Adelaide, SA 5001, Australia

**Keywords:** chemotherapy, mucositis, intestinal microbiota dysbiosis, toll-like receptors

## Abstract

Chemotherapy-induced intestinal mucositis, a painful debilitating condition affecting up to 40–100% of patients undergoing chemotherapy, can reduce the patients’ quality of life, add health care costs and even postpone cancer treatment. In recent years, the relationships between intestinal microbiota dysbiosis and mucositis have drawn much attention in mucositis research. Chemotherapy can shape intestinal microbiota, which, in turn, can aggravate the mucositis through toll-like receptor (TLR) signaling pathways, leading to an increased expression of inflammatory mediators and elevated epithelial cell apoptosis but decreased epithelial cell differentiation and mucosal regeneration. This review summarizes relevant studies related to the relationships of mucositis with chemotherapy regimens, microbiota, TLRs, inflammatory mediators, and intestinal homeostasis, aiming to explore how gut microbiota affects the pathogenesis of mucositis and provides potential new strategies for mucositis alleviation and treatment and development of new therapies.

## 1. Introduction

Cancer is a major global public health burden. Statistically, there were approximately 19,290,000 new cancer cases and 9,960,000 cancer deaths worldwide in 2020 [1]. In recent years, chemotherapy, radiotherapy and surgical operation are still the main modalities for the treatment of cancers despite continuous advances in medical therapies such as targeted therapy and immunotherapy.

During chemotherapy, cytotoxic agents including antimetabolites, alkylating agents, platinum complexes, topoisomerase inhibitors and/or antibiotics are effective at killing cancer cells; however, these agents also indiscriminately affect certain healthy cells, leading to a series of side effects, such as intestinal mucositis, myelosuppression, alopecia, and bone loss [2,3,4,5]. Approximately 40% of patients receiving standard dose chemotherapy and 100% of patients receiving high-dose chemotherapy exhibit nausea, vomiting, abdominal pain, diarrhea and malnutrition associated with intestinal mucositis [6,7,8].

Chemotherapy-induced intestinal mucositis (CIM) not only seriously reduces patients’ quality of life, prolongs hospitalization, adds health care costs, but also impacts adherence to anticancer therapy as it frequently limits the patient’s ability to tolerate treatment, causing schedule delays, interruptions, or premature discontinuation [9]. It has been postulated that CIM occurs in five overlapping phases: initiation, primary damage response, signal amplification, ulceration and healing, and oxidative stress (with production of high levels of reactive oxygen species, ROS), apoptosis and intestinal microbiota dysbiosis (IMD) are closely related to the occurrence and development of CIM [10]. With the continuous technical advancement in microbial species identification through genomic sequencing, the relationship between gut microbiota and CIM has drawn much attention for research into CIM pathogenesis and treatment [11,12,13].

Several studies have revealed that chemotherapy could result in IMD and further aggravate mucositis. However, how chemotherapeutic agents affect gut bacteria and how changed microbiota affect the pathogenesis of CIM are not well documented. In this review, we have summarized relevant studies related to the relationship between gut microbiota and CIM in recent years, aiming to explore how gut microbiota affects the mucosal cellular behavior and CIM pathogenesis and provides potential new strategies for alleviation and treatment of CIM.

## 2. Intestinal Microbiota and Chemotherapy

The intestinal microbial community that inhabits the human gut accounts for about 380 billion microorganisms reaching a biomass of almost 1.8 kg, which are dominated by Firmicutes, Bacteroidetes, Proteobacteria, Actinobacteria, Fusobacteria and Verrucomicrobia [14]. The gut microbiota affects many physiological and pathological processes of the human body such as drug metabolism and vitamin synthesis [15,16]. IMD is not only related to the occurrence and development of cancer, but also affects its therapeutic effect [17,18]. Recent studies have shown that chemotherapeutic drugs can cause IMD, and the intestinal microflora can in turn also affect the effectiveness and toxicity of chemotherapy [19].

### 2.1. Intestinal Microbiota Dysbiosis in Cancer Patients

With the advancement of detection and analysis technology, traditional bacterial culture methods can no longer satisfy researchers’ understanding of microbiota due to their time-consuming nature, high culture requirements, and many influencing factors [20]. The development of molecular biology technology (such as 16S rRNA/DNA sequencing, PCR, molecular probe techniques, gene chip technology and metagenome sequencing) has now provided researchers with a new perspective on gut microbiota [20]. For example, researchers now commonly use 16S rRNA sequencing to detect gut microbiota changes and analyze the microbial community information in multiple samples in a comprehensive and parallel manner, making it possible to study microbiota diversity on a large scale.

Clinical trials have investigated the changes of intestinal microbiota after chemotherapy. Montassier et al. analyzed the alterations of intestinal microbiota of 28 patients with non-Hodgkin’s lymphoma before and after a 5-day myeloablative conditioning regimen with high-dose carmustine, etoposide, aracytine and melphalan through 16S rRNA sequencing analysis. The authors found that the relative abundances of Firmicutes and Actinobacteria in fecal samples were significantly decreased, while that of Proteobacteria significantly increased after the chemotherapy. At the genus level, the relative abundances of Ruminococcus, Oscillospira, Blautia, Lachnospira, Roseburia, Dorea, Coprococcus, Anaerostipes, Clostridium, Collinsella, Adlercreutzia and Bifidobacterium decreased significantly when compared with samples collected before chemotherapy. The authors suggested that IMD was associated with CIM and interventions targeting IMD may be a good strategy to alleviate related complications [21]. Similarly, Motoori et al. quantified the bacteria from patients with esophageal cancer by 16S or 23S rRNA-targeted reverse transcription-quantitative polymerase chain reaction (PCR) and found that the number of Lactobacillus significantly reduced after receiving 5-FU, cisplatin, and docetaxel combined chemotherapy, whereas the number of Clostridium difficile and Enterococci increased significantly. The results indicated that chemotherapy altered intestinal microbiota composition [22].

Recently, Galloway-Peña et al. evaluated microbiota changes of oral swabs and stool samples obtained from 97 acute myeloid leukemia (ALL) patients through 16s rRNA analysis, and the authors showed that microbiota diversity decreased significantly in oral and stool samples; Clostridiales and Blautia in stool samples, Viellonellaceae, Prevotellaceae, and Gemella in oral samples were significantly reduced after chemotherapy. However, Staphylococcus became enriched. The authors suggested that intestinal microbiome evaluation could help with infectious risk stratification to decrease subsequent infectious complications in ALL patients [23]. Rajagopala et al. further established a cohort of 32 pediatric and adolescent ALL patients and 25 healthy siblings to assess the effect of chemotherapy on the gut microbiota. Stool samples were collected every month up to 1 year and detected by 16S rRNA sequencing, and results showed that the microbiota diversity and richness of the ALL group were significantly lower than that of the control group at diagnosis and during chemotherapy, and even species richness remained significantly low after 1 year of treatment [24]. The strength of this study was using healthy sibling controls to compare the gut microbiota in ALL patients undergoing chemotherapy, which controlled for diet, genetics and other related variables. However, due to its small sample size, the findings require additional support and validation.

In a phase II randomized clinical trial on the effectiveness of a probiotic cocktail on oral mucositis in advanced nasopharyngeal cancer patients undergoing concurrent chemoradiotherapy, similar results were found. For instance, the relative abundance of probiotics, such as Lactobacillus, Bifidobacterium and Akkermansia, decreased, while the abundance of harmful bacteria, such as Clostridium, Enterococcus, and Enterobacter increased [25]. A systematic review by Yixia and colleagues summarized that a chemotherapy regimen (including one or more of 5-fluorouracil/FU, capecitabine, oxaliplatin, irinotecan and folinic acid) and surgery may lead to distinct alterations in the gut microbiota composition in colorectal cancer patients particularly in Proteobacteria phylum [26]. Taken together, the body of research shows that chemotherapy usually has a negative effect on intestinal beneficial microbes whereas it has a positive effect on the proliferation of potential pathogenic microorganisms, and exerts a long-term impact on intestinal microflora, which disrupts microbial gut homeostasis for long periods of time after chemotherapy.

### 2.2. Chemotherapeutics Shape Microbiota in Animals

Numerous animal experiments have also evaluated intestinal toxicities and impacts on intestinal microbiota of different chemotherapeutic agents. It has been found that direct or indirect DNA-targeted chemotherapeutic agents often result in severe intestinal toxicities and cause IMD. The main changes of intestinal microflora caused by commonly used chemotherapeutic agents are summarized and shown in Table 1.

Animal experiments largely verified the phenomena observed in clinical trials such as the increased relative abundance of potential pathogenic microorganisms and reduced probiotic ones. Although the reported chemotherapy regimens were quite different, the effects of chemotherapeutic agents on intestinal microflora also showed some similarities, mainly including: (1) the total number and diversity of microbiota in animal feces were generally reduced, especially in those experiments with an earlier sampling time [27,32,33,37,39,40]. (2) Increased abundance of Firmicutes and reduced abundance of Bacteroidetes [31,34,35], although conflicting results had also been reported [30,40,41]; (3) increased abundance of Gram-negative (G-) bacteria, including potential pathogenic microbes such as *E. coli* and Pseudomonas, and decreased abundance of Gram-positive (G+) bacteria, such as Bifidobacterium and Lactobacillus [7,27,28,34,35,36,37,39,41,42,43]. In addition, some intestinal bacteria were found translocated to mesenteric lymph nodes and spleen [38].

## 3. Intestinal Microbiota Dysbiosis Affecting CIM through Toll-like Receptors Signaling Pathways

Toll-like receptors (TLRs) are a family of pattern recognition receptors [44]. They can recognize pathogen-associated molecular patterns on various microbes (such as peptidoglycan, lipopolysaccharide (LPS), lipoteichoic acid, etc.) and the damage-associated molecular patterns from stressed or dying cells [45]. TLRs are expressed throughout the whole intestinal tract by a wide variety of cell types, including intestinal epithelial cells, fibroblasts, macrophages, neutrophils and dendritic cells [46,47]. TLRs molecules are type I transmembrane proteins, which consist of extracellular region, cytoplasmic region, and transmembrane region. The cytoplasmic region is the core region of TLRs, which can cause downstream signaling cascade reactions to stimulate relevant signaling pathways [48].

In general, TLRs could activate the MyD88-dependent pathway (except TLR-3) and the MyD88-independent TRIF/TRAM pathway (TLR-3 and some TLR-4 signals) [45]. MyD88-dependent pathway activation results in the synthesis of proinflammatory cytokines and mediates inflammatory responses. Activation of the MyD88-independent pathway leads to the secretion of type I interferons [47]. In addition, TLR activation also activates other signaling pathways, such as c-Jun N-terminal kinases (JNKs), mitogen-activated protein kinase (MAPKs), interferon regulatory factors (IRFs), extracellular signal- regulated kinases (ERKs), p38, etc. These pathways are essential for orchestrating innate and adaptive immune responses, inflammation, tissue repair, pathogenesis and pathology in the host [47,49].

Under normal physiological conditions, a state of low-grade inflammation maintains due to constant exposure to TLR ligands [50]. Commensal microbiota could decrease NF-κB activation and result in attenuated inflammation, leading to a dynamic balance between pro- and anti-inflammatory responses [51]. However, chemotherapy-induced IMD can disturb TLR signaling pathways and facilitate inflammatory injury [52,53].

### 3.1. Chemotherapeutic Drugs Affecting Expression of TLRs

TLRs have been a major area of focus due to their role as a direct interface between microbial ligands and intestinal cells. Researchers have paid close attention to the effect of different chemotherapeutic agents on the expression of TLRs over the past years. Treatment with 5-FU was found to significantly enhance protein expression of TLR-2 and TLR-4 [27,29,54,55], but slightly reduce the expression of TLR-9 [29]. In addition, the effect on the expression of TLR-2 were different from that of TLR-4. As shown by Justino et al., TLR-2 increased about 1.5 times while TLR-4 increased nearly 4.5 times in the jejunum of mice treated with 5-FU (450 mg/kg) [54]. Using in vitro experimentation, treatment with 5-FU alone was able to promote the expression of TLR-2 and TLR-4 to a similar level in Caco-2 cells; however, in the presence of LPS, 5-FU treatment led to a decrease in TLR-2 and an increase in TLR-4, which suggested that increased G- bacteria might be in favor of the expression of TLR-4 [54].

Unlike 5-FU, Gao et al. [56] found that MTX treatment (20 mg/kg*2d) significantly increased the expression of TLR-2 and TLR-9 in the jejunum of mice, but the effect on TLR-4 was not obvious. However, conflicting results were reported in studies conducted in rats: Hamada et al. [57] found that MTX treatment (15 mg/kg*4d) caused a significant increase in TLR-4 expression in the intestine mucosa, while Sukhotnik et al. [58] observed that MTX administration (20 mg/kg) decreased the expression of TLR-4 in the jejunum.

As shown by Gibson et al., CPT-11 (175 mg/kg) caused no significant changes on the expression of TLR-2, TLR-4, TLR-5 and TLR-9 in jejunum and colon epithelial cells within 120 h in breast cancer-bearing rats; however, the expression of TLR-4 and TLR-5 significantly decreased in jejunal crypts of rats at 96 and 120 h [59]. Similarly, Fikiha et al. observed that there were no significant changes in ileal TLR-2 and TLR-4 proteins within 96 h after CPT-11 (125 mg/kg) treatment [60]. However, Wong et al. observed that when the frequency of CPT-11 treatment (75 mg/kg*4d) increased, the expression of TLR-9 was found enhanced by four times in the ileum of mice, although it still had no obvious effect on TLR-2 [61].

At present, among the 10 TLRs found in the human body, TLR-2 and TLR-4 have been the most widely studied TLRs that are related to intestinal inflammation [62]. Overall, findings from the clinical studies and animal model experiments so far suggest that the effects of chemotherapeutic agents on the expression of TLRs vary significantly, and that the reasons for the variation may be related to the types of drugs, dosages, administration routes and frequencies, sampled organs and animal species.

### 3.2. The Relationship between CIM and TLR-2 Signaling Pathway

In TLR-2 signaling pathway, TLR-2 recognizes conserved molecular patterns associated with G+ bacteria including peptidoglycan and then recruits adaptor proteins MyD88 to initiate the downstream signal cascade (Figure 1). Under normal physiological conditions, besides promoting inflammatory responses, TLR-2 can also induce the expression of the anti-inflammatory cytokine interleukin (IL)-10, thereby inhibiting the excessive activation of inflammatory cells [63], and increasing the expression of P-glycoprotein (P-gp), which actively pumps a broad range of chemically diverse compounds out of the cell to protect the intestinal mucosa [49,64,65]. In addition, TLR-2 activation in epithelial cells can also effectively maintain the integrity of tight junction (TJ) [66]. Thus, activation of TLR-2 may likely play a role to protect the mucosa against some chemotherapeutic drugs. This speculation was supported by Frank et al., who found severe jejunum mucositis, and at the same time, weak expression of P-gp in TLR-2 knockout mice treated with MTX (40 mg/kg*4d). The decreased P-gp gene expression was considered to be the main cause of MTX intestinal damage [64]. However, the protective effects of TLR-2 deletion may be drug class-specific, for example, when treated with CPT-11 (75 mg/kg*4d) [61], or doxorubicin (10 mg/kg) [67], the severity of intestinal damage was found reduced in TLR-2 knockout mice. Moreover, Wu et al. confirmed that 5-FU could aggravate jejunal mucositis through the TLR-2/MyD88/NF-kB signaling pathway [27]. However, the authors observed that numbers of G+ bacteria decreased while those of G- bacteria increased in fecal samples (see Table 1). A similar phenomenon was also observed in rats treated with 5-FU (30 mg/kg*5d), and the plasma LPS level was found to have increased by 10-fold [29]. Thus, chemotherapy-induced intestinal damage seems to be related to TLR-2 activation, the outcome of which on intestinal damage may be drug-specific.

### 3.3. The Relationship between CIM and TLR-4 Signaling

TLR-4 is the major receptor for LPS recognition, which recruits adaptor proteins via the MyD88-dependent pathway and MyD88-independent pathway to activate NF-κB and amplify its effects (Figure 1) [68]. Based on the fact that 5-FU promotes the expression of TLR-4 in mice more strongly than TLR-2 and G- bacteria proliferate in large numbers during chemotherapy, the TLR-4/MyD88/NF-κB/MAPK pathway may play a central role in the intestinal injury [54].

The above point of view is also supported by the fact that the degree of intestinal mucosal damage was positively correlated with the expression levels of TLR-4, IL-6 and tumor necrosis factor-alpha (TNF-α) mRNA in the ileum of 5-FU-treated (200 mg/kg) mice [69]. In MTX-treated (15 mg/kg*4d) rats, Hamada et al. also observed a synchronous increase in TLR-4 and inflammatory cytokines such as TNF-α and IL-1β in intestinal mucosa [57]. On the other hand, a recent study obtained a different result, in which a single intraperitoneal injection of MTX (20 mg/kg) significantly reduced the expression of TLR-4, MyD88 and TRAF6 genes in the jejunum and ileum [58]. However, whether this effect is related with the anti-inflammatory effect of low-dose MTX needs further investigation [70].

## 4. New Perspectives on Pathogenesis of Mucositis and the Non-Intestinal Manifestations of Mucositis Induced by IMD

As early as 2004, Sonis had proposed the “five-phase model” for the pathogenesis of mucositis [10]. Briefly, in the first phase, chemotherapeutic drugs destroy DNA of fast-growing cells, leading to ROS production and cell apoptosis. In the second phase, ROS induces NF-κB activation, which further up-regulates the expression of pro-inflammatory cytokines, leading to the damage of epithelial, endothelial and connective tissues. In the third phase, selected pro-inflammatory cytokines such as TNF-α could positively affect activation of NF-κB, amplify its effect, and aggravate the inflammatory response. In the ulceration stage, due to the suppression of stem cell proliferation and the exhaustion of the mucous layer, the integrity of the mucosa is compromised, and the harmful bacteria colonize, which lead to an ulcer on the alimentary tract, increasing the risk of bacteremia or sepsis. In the healing stage (a few days after stopping chemotherapy), the extracellular matrix releases messenger molecules to promote epithelial cells contiguous to the ulcer to divide, migrate and differentiate into new mucosa, which eventually heals the ulcer.

In this five-phase model, the impact of intestinal microbiota on the incidence of CIM is limited to the ulcer phase. However, as chemotherapy could affect the intestinal microbiota at the early stage [71], we hypothesize that chemotherapy drugs and/or their metabolized products may percolate into intestinal mucosa and kill the commensal bacteria. At the same time, they might also be secreted along with various digestive fluids, directly destroying the bacteria in the gut lumen, and the cell wall components released could be recognized by TLRs, leading to activation of NF-κB. Therefore, the process of chemotherapeutics killing intestinal bacteria to active NF-κB may occur at the same time as the “initiation and message generation” phase of the five-phase model.

On the other hand, the finding of an in vitro study conducted by Vanlancker et al. does not seem to support our proposed possibility as 5-Fluorouracil and irinotecan (SN-38) were found to have limited impact on colon microbial functionality and composition at a dose of 10 mM. However, it is well known that digestive fluids are substantially condensed after being secreted, and the concentration of the chemotherapeutic drugs and/or their metabolized products might be much higher in the gut lumen than in the blood. Thus, further studies are required to investigate these possibilities proposed above.

Moreover, the changed intestinal microbiota may also impact on mucosal cellular behaviors in the healing process. Under normal physiological conditions, epithelial tissues need to undergo perpetual self-renewal, through constant production of new cells that originate from intestinal stem cells (ISCs) residing in the crypts. The daughter cells migrate along the crypt-villus axis, and in the meantime, undergo differentiation towards enterocytes and goblet cells, while Paneth cells move toward the crypt base and intermingle with ISCs. The intestinal mucosa homeostasis is elaborately regulated by several canonical niche pathways, such as Wnt, R-spondins, Notch, Hedgehog, bone morphogenetic protein and YAP/TAZ, which are mainly supplied by the neighboring Paneth cells and subepithelial myofibroblasts. Intriguingly, as an extrinsic ISC niche factor, the role of intestinal microbiota is not restricted to the immune system but can be extended to ISC self-renewal and differentiation [72,73,74]. The decrease in commensal bacteria caused by IMD, especially Bifidobacterium spp. and Lactobacillus spp., are not conducive to healing and regeneration of intestinal epithelium as a result of reduced production of lactate (derived from the bacteria and serving as a substrate for oxidative phosphorylation of ISCs supporting the function of ISCs) [75]. Conversely, supplementation with lactate-producing bacteria could promote cell proliferation and differentiation [76]. Furthermore, the healing process would also be postponed because extracellular matrix, being vital to stem cell proliferation and subsequent differentiation, could be destroyed by over-expressed MMPs upon bacterial infection [72,77].

Another obvious effect of the intestinal microbiota may occur after the mucosal healing stage. Studies have shown that the intestinal microbiota has not returned to a stable state even a long time after the mucosal healing, and this is usually closely related to another side effect of chemotherapy, namely cognitive impairment induced by chemotherapy (CICI). CICI is also a complication of chemotherapy, which affects processing speed, memory, executive function, learning and concentration, and may seriously affect the quality of life. However, the pathogenesis for CICI has not yet been fully elucidated [78].

In recent years, the immunomodulatory properties of the intestinal microbiota and its ability to control neuroinflammation have been thought to play a role in CICI through the gut-brain axis [79]. Gut-brain axis is the connection which exists between the central nervous system and the alimentary tract, in which the microbiota exerts a profound effect on the central nervous system, affecting behavioral, emotional and cognitive domains [80], potentially through the possible mechanisms as follows: (1) Short-chain fatty acids, which are produced by bacterial fermentation, can regulate the central nervous system, innate immunity and blood-brain barrier permeability [81]. (2) Microbes can send signals to the vagal afferents to affect memory [82,83]. (3) Changes in microbial homeostasis may affect the cognitive function by disrupting the hypothalamic-pituitary-adrenal axis and changing the stress circuit [84]. The pathobiology of CICI and risk factors have been recently summarized by Subramaniam et al. [85].

In addition, “fear of cancer recurrence (FCR)” is also believed to be related to the microbial dysbiosis. In a clinical study, Okubo et al. observed that a lower bacterial diversity, an increased relative abundance of Bacteroidetes and a decreased abundance of Firmicutes were significantly related to FCR in breast cancer survivors who had received chemotherapy, when compared to those with no such history. The authors thought that abnormal changes in microbiota might increase intestinal barrier permeability and inflammation, which leading to the dysfunctional processing of fear memory to FCR [86].

Taken together, gut microbial dysbiosis plays an important role in the pathogenesis of mucositis and further affects the mental health of patients/survivors through the microbiota-gut-brain axis.

## 5. Prophylactic Effect of Antibiotics, Probiotics and Natural Products

The recognition of the role of intestinal microbial dysbiosis in the pathogenesis of chemotherapy-induced mucositis may aid the development/use of agents/strategies to modulate chemotherapy-induced intestinal damage. Clinical practice guidelines recommend the use of antimicrobial agents in neutropenic patients with cancer [87], and several previous studies also proved that taking antibiotics during CPT-11 chemotherapy could alleviate the delayed diarrhea induced by CPT-11 (due to the toxic effect of its active metabolite SN-38 to the bacteria in the colon) [88,89]. In a recent study, Ziegler et al. compared effects of broad-spectrum antibiotics and levofloxacin (250 mg/day) prophylaxis (from day 8 after chemotherapy until engraftment or neutropenic fever) on the bloodstream infections in patients with hematologic malignancy, and the results showed that levofloxacin prophylaxis altered gut microbiota to a lesser extent (without affecting microbiota diversity), while broad-spectrum beta-lactam antibiotics were associated with decreased alpha diversity [90]. However, the effects of the timing and duration of antibiotic exposures remain to be explored. Although antibiotics are widely used in chemotherapeutic regimens to reduce the risk of sepsis or infections in cancer patients, their impact on the gut microbiota (which are related to the types of antibiotics) have been of concern. In addition, some concerns have been raised that both chemotherapeutic drugs and antibiotics can induce drug-resistant strains of microbiota, which may further increase the risk of death from infections [91]. Nonetheless, antibiotic prophylaxis is still encouraged based on existing clinical evidence [92].

In the last two decades, a large number of studies have demonstrated the potential of probiotics to ameliorate toxicities of chemotherapeutic agents and have attested the usefulness of probiotics in association with cancer chemotherapy. As reviewed by Badgeley et al., supplementation with probiotics has several benefits for cancer patients such as enhancing cancer cell apoptosis, reducing chemotherapy toxicity and modulating host immune response [93]. A phase II randomized clinical trial recently conducted by Xia et al. confirmed that a probiotic cocktail can reduce concurrent chemoradiotherapy-induced oral mucositis in nasopharyngeal cancer patients [25]. Interestingly, fecal microbiota transplantation could be used to treat severe Clostridioides difficile colitis in a pediatric patient with non-Hodgkin lymphoma [94]. By animal experimentation, De Jesus et al. found that in BALB/c mice treated with 5-FU (300 mg/kg), prophylactic administration of Lactobacillus delbrueckii subsp. lactis CIDCA 133 significantly alleviated the intestinal mucositis, and the beneficial effects were manifested by lower body weight loss, normal intestinal length, and reduced inflammatory infiltration and intestinal permeability [95]. Similarly, oral administration of probiotics Lactobacillus casei variety rhamnosus, Lactobacillus acidophilus or Bifidobacterium bifidum was found to be a safe therapeutic option for immunodeficiency mice to alleviate 5-FU (30 mg/kg*5d)-induced intestinal mucositis and bacteremia in the blood [96]. In rats treated with oxaliplatin (OXL) (8 mg/kg*3d) and 5-FU (75 mg/kg*3d), prophylactic treatment of Bifidobacterium infantis ameliorated the intestinal mucositis [97]. As for potential mechanisms for the protective action of probiotics, prophylactic treatment of B. infantis was found to decrease expression levels of proinflammatory cytokines (IL-6, IL-1β, and TNF-α) in the intestine of rats treated with OXL (8 mg/kg*3d) and 5-FU (75 mg/kg*3d) [97]. Similarly, treatment with Saccharomyces boulardii CNCM I-745 inhibited the expression of TLR-2, TLR-4, MyD88, NF-κB, ERK1/2, phospho-p38, phospho-JNK, TNF-α, IL-1β and C-X-C motif chemokine ligand 1 (CXCL-1) in the jejunum/ileum of mice treated with 5-FU (450 mg/kg) [54]. These results suggest that probiotics or at least CNCM I-745 can modulate TLRs/MyD88/NF-κB/MAPK signaling pathways to inhibit the occurrence of mucositis [54]. In addition, lipoteichoic acid from Lactobacillus rhamnosus GG (LGG) could bind to TLR-2 on pericryptal macrophages to protect ISCs from radiation injury [98]. In TLR-2 or MyD88 knockout mice, the protective effect of LGG on epithelial cells was lost, which verified that the beneficial effect of this probiotic is correlated with TLRs signaling [99]. In addition, Lactobacilli could enhance expression of tight junction proteins and Faecalibacterium prausnitzii could produce butyrate to fight against inflammation, decrease intestinal permeability and stimulate immune activation [100,101].

Taken together, results from most studies support the conclusion that the probiotics can reduce the incidence and mitigate the severity of CIM. However, some potential limitations should be mentioned, including potential local or systemic infections caused by the probiotics, metabolic abnormalities, excessive immune stimulation and some strains that may carry potential antibiotic resistance genes, which may increase the risk in immunocompromised patients [99,102].

In addition, in recent years, natural products including Chinese herbal medicines have displayed distinct clinical outcomes in managing chemotherapy-induced side effects including gut mucositis, nausea and vomiting [103,104,105], and some of them have been verified in animal experiments, such as Wuzi Yanzong Pills [106], Shengjiang Xiexin decoction [107], Gegen Qinlian decoction [108], Qingjie Fuzheng Granule [109], Amomi Fructus [110], Steamed rehmannia root [111], Patchouli [27,112], Saikosaponin-A [7], Aquilaria agallocha [113] and Andrographolide [114], with the underlying mechanisms of action involving antioxidation, anti-inflammation and anti-apoptosis. In addition, some ingredients extracted from *Crassostrea hongkongensis* [55], *Dendrobium sonia* [115], *Dendrobium huoshanense* [116], *Schisandrae chinensis* [117], *Salvia miltiorrhiza* [118], *Scutellaria baicalensis* [119], and *Pogostemon cablin* [27] are likely to regulate gut microbiota through promoting the proliferation of commensal microbiota and inhibiting pathogenic bacteria. Furthermore, Patchouli alcohol extract and Patchouli oil have been proved to alleviate CIM by modulating TLR signaling pathways [27,112]. Although natural products and probiotics have the activity of mitigating mucositis, their effects on the efficacy of chemotherapy still need to be further investigated.

## 6. Conclusions and Further Perspectives

CIM is one of the most commonly occurring adverse effects of chemotherapy. Clinical studies have proved that CIM is accompanied by IMD, which has been partly confirmed by animal experiments. Although the chemotherapy regimens vary greatly with chemotherapeutic agents, administration routes, dosages and times, and animal species or strains, and sampling time, significant alterations in gut microbiota have been observed after chemotherapy.

The present data have shown that chemotherapy-induced IMD can be mainly characterized by reduced diversity and total numbers of intestinal bacteria, reduced G+ and increased G- bacteria, and reduced beneficial bacteria and increased potentially pathogenic ones. Chemotherapy-induced IMD has been shown to adversely impact on intestinal stem cells and mucosa homeostasis. In addition, chemotherapy leads to the alterations in the expression of members of TLR signaling pathways, especially those related with TLR-2 and TLR-4 pathways resulting in increased expression of inflammatory mediators in the mucosa. Thus, it can be concluded that chemotherapy can induce IMD, which in turn can aggravate mucositis via interaction with TLRs.

However, how chemotherapeutic agents affect intestinal bacteria largely remains unknown. In addition, while multiple drugs are usually used in cancer treatment, many of these drugs have not been evaluated together for their effects on intestinal microbiota and CIM using animal models, which needs urgent attention. Furthermore, due to certain limitations in microbiological research such as individual difference, underlying diseases, sampling accuracy, and disadvantage of the sequencing method itself, controversy over the accuracy of the microbial data also exists. Lastly, since positive effects of probiotics and natural products on CIM have been achieved clinically and experimentally, it is worthy of further focusing on their research and development, especially those of natural products or derivatives due to their multi-target therapeutic effects.

## Figures and Tables

**Figure 1 ijms-22-09474-f001:**
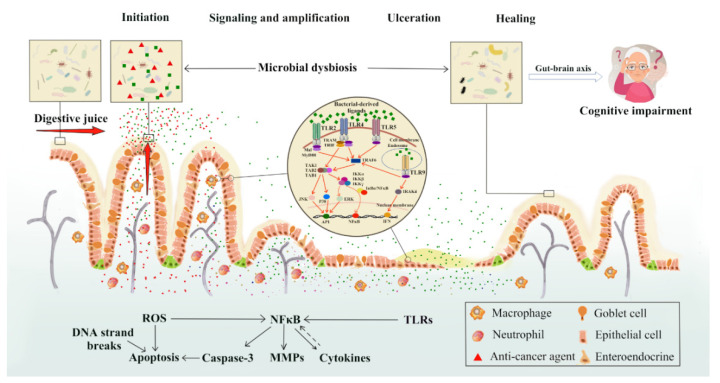
New perspectives on impacts of chemotherapy-induced intestinal microbial dysbiosis on pathogenesis of mucositis and the non-intestinal manifestations of mucositis. On the basis of the five-phase model for chemotherapy-induced mucositis put forward by Sonis in 2004, this review proposed the possible roles of intestinal microbial dysbiosis at various phases of mucositis pathogenesis and in the cognitive impairment of patients/survivors. Chemotherapy drugs or their metabolites may percolate into intestinal mucosa and may also be secreted along with various digestive fluids, directly destroying the bacteria in the gut lumen, altering their total numbers and compositions. Cell wall components of the killed bacteria are recognized by toll-like receptors (TLRs), such as peptidoglycan through TLR-2, lipopolysaccharide (LPS) through TLR-4, flagelin through TLR-5 and non-methylated DNA (CpG DNA) through TLR-9. All TLRs recruit adaptor proteins via the myeloid differentiation factor 88 (MyD88)-dependent pathway and/or adaptor-inducing interferon-β (TRIF)-dependent pathway to activate NF-κB and/or ERK/JNK/p38 kinases, which further lead to increased expression of inflammatory mediators. Intestinal microbial dysbiosis may further affect the mental health of patients/survivors through the microbiota-gut-brain axis. MAL: MyD88-adaptor like protein; IRAK: interleukin-1 receptor associated kinase; TRAF6: TNF-α receptor association factor 6; TAK: thylakoid associated kinase; TAB: TAK-binding protein; NF-κB: nuclear factor-κB; IκB: NF-κB inhibitor; IKK: IκB kinase; ERK: extracellular signal-regulated kinase; JNK: c-Jun N-terminal kinase; AP-1: activator protein 1; IFN: interferon.

**Table 1 ijms-22-09474-t001:** Effects of different chemotherapeutic regimens on intestinal microbiota in animal experiments.

Chemotherapy Regimen	Effects on Gut Microbiota	Reference
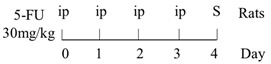	Diversity of the microbiota community↓;Bifidobacterium, Lactobacillus↓;Bacteroides, Escherichia, Helicobacter, Parabacteroides↑.	[27]
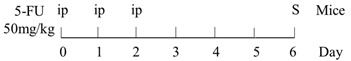	E. coli↑;Lactobacillus spp↓.	[7]
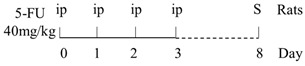	Lactobacillus, Prevotella↓;Bacteroides, Escherichia↑.	[28]
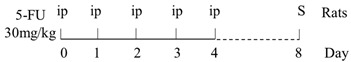	Total bacteria abundance↓;Lactobacillus, Clostridium cluster III and XIVa↓;Enterococcus spp., Lachnospiraceae↑.	[29]
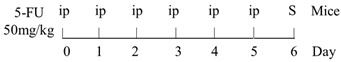	Firmicutes, Actinobacteria, Tenericutes↓;Bacteroidetes, Proteobacteria↑.	[30]
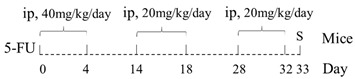	Firmicutes, Verrucomicrobia↑;Bacteroidetes↓.	[31]
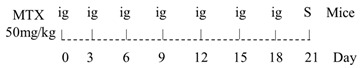	Diversity of the microbiota community↓;Ruminococcaceae, Eisenbergiella and Marvinbryantia↑.	[32]
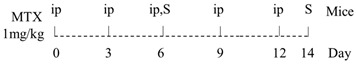	On Day6 and Day14, diversity of microbiota community↓;Ruminococcaceae, Bacteroides fragilis ↓; Lachnospiraceae↑.	[33]
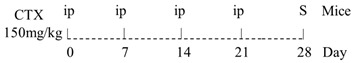	Firmicutes, Tenericutes↑;Bacteroidetes, Cyanobacteria↓.	[34]
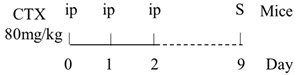	Firmicutes, Proteobacteria↑;Bacteroidetes, Lactobacillus, Bifidobacterium↓.	[35]
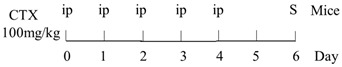	E. coli, Pseudomonas, Enterococcus↑.	[36]
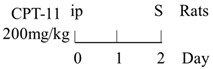	Diversity of the microbiota community↓;Bacteroidetes, Actinobacteria↓;Fusobacteria, Proteobacteria↑.	[37]
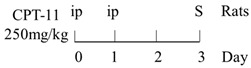	Bacteria were detected in mesenteric lymph node and spleen.	[38]
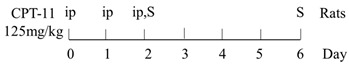	The total bacteria number on Day2↓;Lactobacillus on Day2, Bifidobacterium on Day2 and Day6↓.	[39]
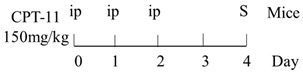	Richness of microbiota↓; Bacteroidetes and Firmicutes not changed;Proteobacteria, Porphyromonadaceae, Mogibacteriaceae↑;Rikenellaceae↓.	[40]
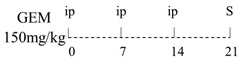	Bacteroidetes, Firmicutes↓; Verrucomicrobia, Proteobacteria, E. coli↑;Akkermansia muciniphila and Peptoclostridium difficile were detected.	[41]
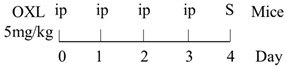	Bacteroidetes↓;Protebacteria↑.	[42]

↑, increases; ↓, decreases.

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
