# Peer review of "Chemotherapy-Induced Intestinal Microbiota Dysbiosis Impairs Mucosal Homeostasis by Modulating Toll-like Receptor Signaling Pathways"

_ijms, 2021, doi:10.3390/ijms22179474_

Round 1
Reviewer 1 Report
The manuscript was reviewed for publication in the journal. The manuscript was designed to review relevant studies related to the relationship between gut microbiota and chemotherapy-induced intestinal mucositis (CIM) in recent years. It is the reviewer’s opinion that the manuscript is quite interesting and easy to follow for the readers. However, it appears that there are a couple of concerns in the manuscript.
1) The main changes of intestinal microflora caused by commonly used chemotherapeutic agents were summarized and shown in Table 1. These data contained only animal experiments. How about the reports in human? The authors should show the data or discuss the issue.
2) The authors showed the review regarding the relationship between gut microbiota and CIM in recent years. There are several studies regarding the topic in 2020 and 2021. The authors should add the recent literatures and discuss the topic.
3) The authors focused toll-like receptor signaling pathway regarding the relationship between gut microbiota and CIM. How about other signaling pathways? The authors should discuss the issue.
4) The title of Table 1 is A summary of microRNAs and their targets involved in osteogenesis and adipogenesis. This may be wrong? The authors should fix the point.
5) Minors
No space before square bracket. For example, 2020[1] to 2020 [1].
No space before units. For example, 30mg/kg to 30 mg/kg.
Author Response
ijms-1355589 Responses to Reviewer1 Comments
Dear Editors and Reviewers,
We would like to thank you for the opportunity for us to revise this work. We highly appreciated the expert reviewers’ insightful and constructive comments and suggestions for this work. The manuscript has been revised based on your comments and suggestions.
Reviewer #1
1) The main changes of intestinal microflora caused by commonly used chemotherapeutic agents were summarized and shown in Table 1. These data contained only animal experiments. How about the reports in human? The authors should show the data or discuss the issue.
Response: Thank you for your suggestion. The clinical reports on the changes of intestinal microflora are now shown and discussed in Section 2.1.
2) The authors showed the review regarding the relationship between gut microbiota and CIM in recent years. There are several studies regarding the topic in 2020 and 2021. The authors should add the recent literatures and discuss the topic.
Response: Thank you for your suggestion. The new reports on this topic have been searched again in the database of Web of Science-SCI/SSCI/A&HCI/CPCI, and 4 more references have now been cited in the revised manuscript.
Xia C, Jiang C, Li W, Wei J, Hong H, Li J, Feng L, Wei H, Xin H and Chen T (2021) A Phase II Randomized Clinical Trial and Mechanistic Studies Using Improved Probiotics to Prevent Oral Mucositis Induced by Concurrent Radiotherapy and Chemotherapy in Nasopharyngeal Carcinoma. Front. Immunol. 12:618150
Sauruk da Silva K, Carla da Silveira B, Bueno LR, Malaquias da Silva LC, da Silva Fonseca L, Fernandes ES and Maria-Ferreira D (2021) Beneficial Effects of Polysaccharides on the Epithelial Barrier Function in Intestinal Mucositis. Front. Physiol. 12:714846.
Vanlancker E, Vanhoecke B, Stringer A and Van de Wiele T. 5-Fluorouracil and irinotecan (SN-38) have limited impact on colon microbial functionality and composition in vitro. PeerJ 2017; 5:e4017.
Sabus A, Merrow M, Heiden A, Boster J, Koo J, Franklin ARK. Fecal Microbiota Transplantation for Treatment of Severe Clostridioides difficile Colitis in a Pediatric Patient With Non-Hodgkin Lymphoma. J Pediatr Hematol Oncol 2021;43:e897–e899
3) The authors focused toll-like receptor signaling pathway regarding the relationship between gut microbiota and CIM. How about other signaling pathways? The authors should discuss the issue.
Response: Thank you for your comments. As this review mainly focuses on TLR signaling pathway in the relationship between gut microbiota and CIM, the other signaling pathways related are only briefly discussed in Section 3 and Section 4.
4) The title of Table 1 is A summary of microRNAs and their targets involved in osteogenesis and adipogenesis. This may be wrong? The authors should fix the point.
Response: Thank you for pointing this out. This mistake has now been corrected. New title: “Table 1. Effects of different chemotherapeutic regimens on intestinal microbiota in animal experiments”.
5) Minors
No space before square bracket. For example, 2020[1] to 2020 [1].
No space before units. For example, 30mg/kg to 30 mg/kg.
Response: Thank you for pointing these out. These and similar errors have been corrected in the revised manuscript.

Reviewer 2 Report
Research Summary:
The authors present a review of research on chemotherapy related gut microbiome dysbiosis, specific effects on TLRs, and infer a broader impact of microbiome dysbiosis on chemotherapy treatment as well as posited temporal interaction throughout treatment. They propose additional attention to gut microbiome/chemotherapy treatment dynamics that may lead to enhanced treatment for mitigation of side effects.
General perception:
In terms of content, this is a generally well written review and important to the field. The initial jump from chemo related dysbiosis to TLRs felt disconnected and abrupt but the authors summarize the entire story well later in the manuscript. Language was coherent and understandable. However, there is a need for additional English language grammar and structure work on the manuscript before it is ready for publication.
Detailed comments and questions on sections:
General:
- General proofreading for grammar and language. I have detailed a number of changes below but the list is not comprehensive.
Figure 1:
- Figure seems appropriate.
Table 1:
- This is a massive table as shown. I recommend trying to get it to fit on a single page.
Abstract:
- “has summarized” to “summarizes”
- “provide potential” to “provides potential”
Introduction:
Pg 1 “Approximate 19,290,000” to ““Approximately 19,290,000”
Pg 1“During chemotherapy, cytotoxic agents including antimetabolites, alkylating agents, platinum complexes, topoisomerase inhibitors and/or antibiotics are effective at killing cancer cell.” Seems to warrant a citation.
Pg 1 “and may also reduce the dosages of chemotherapeutics” I would clarify this better that it reduces the deliverable dosages of chemotherapeutics below the needed or recommended dosages. As written, it seems to imply a reduction in NEEDED chemo.
Pg 2 “Amounts of studies revealed” to “Several studies have revealed”
Intestinal microbiota and chemotherapy:
Pg 2 “inhabits the human gut”
Pg 2 “the weight of almost 1.8kg” to “a biomass of almost 1.8kg”
Pg 2 “Recent studies revealed” to “Recent studies have shown”
Pg 2 “Clinical trials have investigated “
Pg 2 “analysis, the authors found” to “analysis. The authors found”
Pg 3 “chemotherapy, however, Staphylococcus was enriched.” to “chemotherapy. However, Staphylococcus was enriched.”
Pg 3 “The author thought” to “The authors suggested that”
Pg 3 “gut microbiota, stool samples” to “gut microbiota. Stool samples”
Pg 3 “variables, however, due to its” to “variables. However, due to its”
Pg 3 “due to its small sample size, the results are not convincing enough.” to “due to its small sample size, the findings require additional support and validation.”
Pg 3 “Taken together, the body of research shows that chemotherapy exerts a long-term impact on intestinal microflora, which disrupts microbial gut homeostasis for long periods of time after chemotherapy.
Pg 3 “Numerous animal experiments have also evaluated”
Pg 5 “The animal experiments have” to “Animal experiments”
Pg 5 “although different results had also been reported” to “although conflicting results have also been reported”
Pg 5 “Overall, the animal experiments have largely verified the phenomena observed in clinical studies such as the increased relative abundances of potentially pathogenic microorganisms and reduced probiotic ones.” This was already stated at the beginning of this paragraph. Suggest re-write or removal.
Intestinal microbiota dysbiosis affecting CIM through Toll-like receptors (TLRs) signaling pathways:
Pg 5 TLR should be spelled out in its first usage in the text, not just the section title.
Pg 5 “TLRs are expressed”
Pg 5 “TLRs molecules areⅠtype” to “TLRs molecules are type I”
Pg 6 “Using in vitro experimentation, treatment with 5-FU alone was able to promote the expression of TLR-2 and TLR-4 to a similar level in Caco-2 cells”
I am going to stop detailing grammar and language needs at this point. I strongly recommend the authors review the entire text for these types of minor language and grammar needs.
New perspectives on pathogenesis of mucositis and the non-intestinal manifesta- tions of mucositis induced by IMD
- No major comments on content.
Prophylactic effect of antibiotics, probiotics and natural products
- No major comments on content.
Conclusion and further perspectives
- Conclusion is pretty well written and sums up the story of the review as well as making a good call to action for future work.
Round 2
Reviewer 1 Report
The manuscript was reviewed for publication in the journal. The manuscript was designed to review relevant studies related to the relationship between gut microbiota and chemotherapy-induced intestinal mucositis (CIM) in recent years. It is the reviewer’s opinion that the manuscript is quite interesting and easy to follow for the readers. The authors have promptly fixed the manuscript and explained/discussed all issues suggested. I have no more concern in the manuscript.